# MetaDragonBoat: Exploring Paddling Techniques of Virtual Dragon Boating in a Metaverse Campus

## ABSTRACT

The preservation of cultural heritage, as mandated by the United Nations Sustainable Development Goals (SDGs), is integral to sustainable urban development. This paper focuses on the Dragon Boat Festival, a prominent event in Chinese cultural heritage, and proposes leveraging immersive technologies, particularly Virtual Reality (VR), to enhance its preservation and accessibility. Traditionally, participation in the festival's dragon boat races was limited to elite athletes, excluding broader demographics. Our proposed solution, named MetaDragonBoat, enables virtual participation in dragon boat racing, offering immersive experiences that replicate physical exertion through a cultural journey. Thus, we build a digital twin of a university campus located in a region with a rich dragon boat racing tradition. Coupled with three paddling techniques that are enabled by either commercial controllers or physical paddle controllers with haptic feedback, diversified users can engage in realistic rowing experiences. Our results demonstrate that by integrating resistance into the paddle controls, users could simulate the physical effort of dragon boat racing, promoting a deeper understanding and appreciation of this cultural heritage.

## CCS CONCEPTS

• **Human-centered computing** → **Virtual reality**; **Empirical studies in interaction design**; • **Hardware** → **Haptic devices**; Electro-mechanical devices.

## KEYWORDS

Culture and Art, Metaverse, Exergame, Haptic Simulator

**ACM Reference Format:**
Anonymous Author(s). 2024. MetaDragonBoat: Exploring Paddling Techniques of Virtual Dragon Boating in a Metaverse Campus. In *Proceedings of Proceedings of the 32nd ACM International Conference on Multimedia (ACM MM '24)*. ACM, New York, NY, USA, 10 pages. https://doi.org/XXXXXXX.XXXXXXX

## 1 INTRODUCTION

The United Nations Sustainable Development Goals (SDG)[1] underscore the critical need to protect and safeguard the globe's cultural heritage. This overarching directive emphasizes the intricate connection between the perception of intangible festival heritage and

[1]https://www.un.org/sustainabledevelopment/cities/

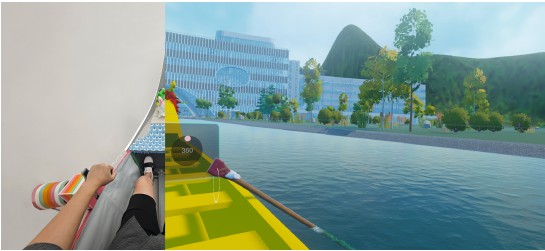

**Figure 1: A digital twin of physical and virtual dragon boat.**

the broader vistas of urban and architectural heritage. Such a linkage is pivotal in crafting a cultural management framework that fosters sustainable urban development [1]. Within this context, the Dragon Boat Festival emerges as a quintessential exemplar of Chinese cultural heritage, garnering international acclaim for its rich cultural tapestry. Held annually in various Chinese cities, including Guangdong, Guangxi and Hainan, this festival transcends its roots as a community and elite sports event to offer a captivating tourism experience that draws global audiences [13]. However, traditionally, the visceral thrill and communal bond of dragon boat racing were exclusive realms of elite sportsmen. This paper posits that through immersive technologies, a broader demographic can virtually partake in the Dragon Boat Festival, thereby enhancing the preservation of this invaluable cultural heritage.

Although immersive technologies have been applied in various forms of athlete training [37, 43] and demonstrating cultural sites [14, 15, 49], our work uniquely considers multimedia experiences of the footage of cultural heritage through the rigorous and proactive engagements on a virtual dragon boat. The dragon boat race, which originated in the Lingnan region of China, is a kind of devotion to the deity since participants believe that the deity takes pleasure in seeing their *exertion* throughout the competition [66]. Therefore, it is essential to do the utmost to replicate the physical actions by studying interactive systems that operate on the virtual dragon boat. In addition, a digital twin of a university campus situated in the Lingnan area, which is in close proximity to a renowned scenic river for dragon boat racing, has been constructed to provide users with authentic yet immersive *experiences*.

Thus, we designed and implemented a metaverse solution, *MetaDragonBoat*, to tackle the issues outlined before and provide physical exertion and immersive experiences. Our solution consists of two major components: the virtual dragon boat in the digital twins (Figure 1) and three paddling techniques. Among the techniques, we design and build a pair of physical paddle controllers with haptic feedback that emulate the actions of physical dragon boat rowing and water resistance. Within such a digital twin and hardware configuration, users with VR headsets can do dragon boat rowing, which entails physical exertion [3]. This leads to a realistic cultural experience around the Lingnan River. It is important to note that by

                                                        

integrating resistance into the paddle controls, users can mimic the physical effort involved in actual dragon boat racing and, hence, the rowing experience. Specifically, this paper examines three paddling techniques, including the physical paddle controllers and the other two methods enabled by standard operations on the VR controllers, driven by joysticks [9] or Inertial Measurement Units (IMU) [2].

A total of 18 participants were recruited, and they were told to complete a water trail on the virtual dragon boat, under the aforementioned three methods. We measured the participants' heart rate, user experiences, workload, and motion sickness [20, 25, 50, 64, 65], to evaluate the experience and exertion offered by the metaverse solution. As a result, the participants with the physical paddle controllers achieved significantly higher heart rates and workloads than the counterparts of standard commercial controllers. Moreover, the participants reflect a significant difference in user experience regarding pragmatic quality between the methods. Furthermore, despite the newly proposed hardware introducing more rigorous motions than the standard methods, it did not impact motion sickness. Overall, our solution contains virtual dragon boats and hardware configuration, delivering a satisfactory experience with a sense of liveliness, aligned with the objectives of preserving cultural heritage and fostering spiritual dedication outlined previously.

The paper's contribution is primarily threefold under the intersection of immersive technologies and cultural heritage. First, we built a physical mock-up of a dragon boat with aesthetic designs. This mock-up mirrors a virtual dragon boat inside a virtual campus, enabling amateur users to have a cultural experience from the comfort of an indoor environment. Second, the proposed hardware interaction techniques, supported by haptic feedback, balance user experience and exertion, are characterized by a reasonable increase in workload and physical activities. Third, our user evaluation reveals that our system and newly proposed hardware device meticulously introduce exertion to the participants while reserving the usability and utility with comparable levels of simulator sickness as the standard methods.

## 2 RELATED WORK

***Metaverse as multimedia application***. As consumer-level extended reality technologies are widely available, metaverse applications become more accessible than ever before. Virtual environments have appeared in recent years. They serve various purposes due to the enrich experiences, such as digital museums [24, 53], education [10], training [22, 29], sensory stimulation [40, 41], authentication [30], meditation [60], and gaming [31, 65]. For instance, the multimedia community leverages the metaverse as the medium to promote lighting due to the security of the teaching venues and the relevant costs [61]. Another recent example built a virtual low-poly style campus for those not physically accessible to a campus tour [12]. However, studies rarely take advantage of metaverse to experience cultural heritage, let alone use enriched feedback for user experience and exertion. Our work serves as a first effort to deliver dragon boat experiences to various users beyond the elite sportsman, to ensure an all-encompassing approach to promote the cultural heritage.

***Virtual reality for sport activities***. Integrating multimedia into social sports, propelled by Mueller et al.'s exertion framework

[33], has significantly influenced the community, emphasizing the blend of physical engagement and interaction techniques in sports technology [34, 35]. This approach has revolutionized indoor endurance training, notably with the introduction of Zwift[2] in 2014, which brought cycling and running into immersive virtual environments for a more engaging workout experience. Following these advancements, Neumann et al. [38] reviewed VR's role in endurance sports, proposing a conceptual framework for future VR sports, with a focus primarily on cycling and less on rowing. Rowing training with multimedia applications opens opportunities for training athletes, and thus usually focuses on physiological characteristics, such as oxygen uptake, carbon dioxide production, ventilation, and completion time [23]. Sebastian et al. [3] further explored VR's impact on rowing, highlighting how VR can enhance athletes' racing skills and workout experiences, with the measurements of breath, strokes per minute, rhythm, stroke length and movement, as well as completion time. In contrast, our work uniquely considers non-professional players and thus strikes a balance between user experience and exertion, offering an alternative perspective on the development of aquatic sports activities in VR.

***Haptic Interaction Techniques***. Interaction techniques for haptic feedback have long been a focus of multimedia research [8, 18, 19, 58] for diversified applications [56], due to the goals of creating realistic virtual environments [4, 46] and increased utility [6]. The traditional work begins with techniques for more fluid and natural interactions with 3D objects through offering forces once users interact with virtual instances [28, 48]. Subsequently, the interaction design of haptic feedback is not limited to haptic gloves [5, 32, 54] and attached surfaces on users [11, 46]. Diversified purposes, e.g. deformable object manipulation, emulating objects of irregular shapes, energy harvesting and user redirection [7, 27, 47, 55], have been demonstrated by recent work. In the context of rowing, the combination of visual, audio, and haptic feedback can improve the athletes' performance [45, 52]. One latest work, named VR4VRT, contains haptic gloves and motion trackers to investigate rowing techniques on a standard rowing machine [59]. Their preliminary findings show the relationship between velocity and trajectory using a variety of visual, audio, and haptic cues. In another four-person collaborative setting [57], haptic feedback is enabled by robotic arms to emulate water resistance encountered by the water paddle of the virtual world, in which the teamwork was evaluated by measuring the synchronisation of the four crews' rowing motions and the trajectory of the boat. In contrast, our research presents an electromechanical device that integrates resistance with haptic feedback. This device enables novice users to simulate the sensation of water resistance while interacting with virtual paddles in a water-like environment.

## 3 SYSTEM IMPLEMENTATION

To promote the dragon boat culture through virtual worlds and digital twins, our system offers an immersive replication of the real-world rowing boat experience characterized by a sense of presence and realism. We build a dragon boat mock-up (Figure 2) that connects to the system containing three major components: (a)

---

[2]https://www.zwift.com/

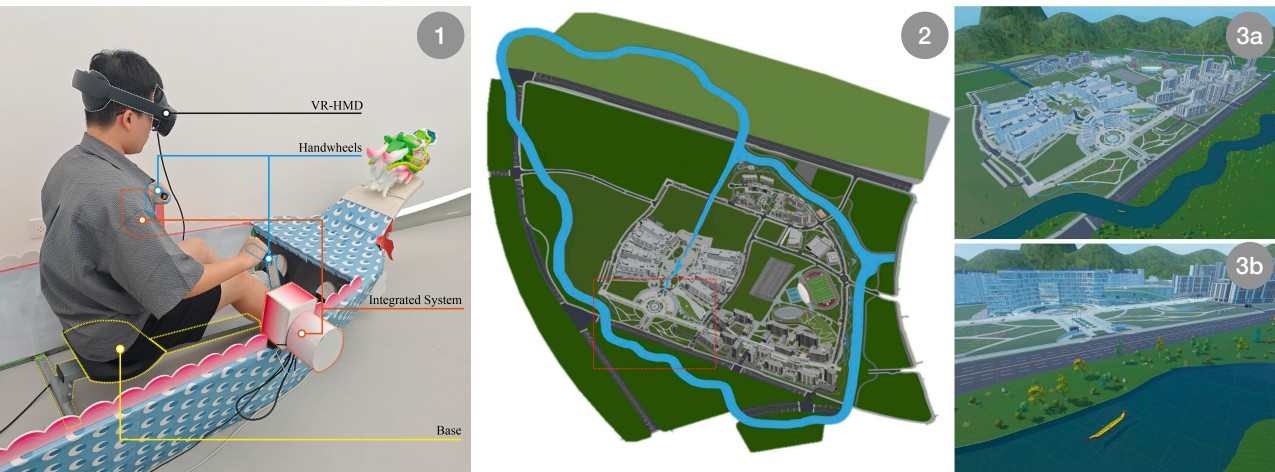

**Figure 2: The digital twin of the dragon boat and the virtual campus adjacent to scenic rivers in China's Lingnan region. (1) Head-mounted display (HMD) users interacting with the dragon boat mock-up, while users equipped with our proposed hardware simulate rowing the boat; (2) A bird's-eye view of the virtual campus; (3a) A zoom-in view from the red-colour box in (2); and (3b) a further zoom-in reveals the yellow-coloured dragon boat.**

A digital twin campus and virtual dragon boats, (b) Three paddling techniques for boat movement in the virtual world for diversified users (Section 3.2), supported by either the headset's handheld controllers or (c) a newly proposed hardware device attached to the dragon boat mock-up (Section 3.3).

## 3.1 Virtual Environments and Interfaces

To offer users a vivid experience of dragon boat rowing, the renowned scenic rivers in China's Lingnan region inspire our metaverse campus. Thus, we utilise the *Unity* game engine that specializes in producing realistic virtual worlds in developing the virtual environment. For cultural heritage preservation and engaging more audiences in the dragon boat racing tradition, our virtual environment is a digital twin of a university campus, consisting of a cluster of buildings encompassed by the waterway (Figure 2 (2–3)). To ensure a smooth experience on head-mounted displays (HMDs), we trim the complicated features, and thus, the 3D model is lightweight.

This virtual architecture serves not only as a visual and interactive element within the VR environment but also functions as an educational tool that aligns with the goal of cultural heritage conservation. As such, users experience a one-kilometre-long virtual reality (VR) dragon boat journey. The virtual environment supports up to six separate lanes, each approximately 13.5 meters in width, making up the racing track. Additionally, green and red lines indicate the beginning and ending locations, respectively. Two important lines are included: a green line that marks the beginning of the race is activated when the boat's bow exceeds this threshold, and a red line that marks the finish line is indicated by the boat's tail crossing it (Figure 7). To visualize the rowing distance, the lanes are marked with buoys positioned at 10-meter intervals.

## 3.2 Paddling Techniques for Boat Movements

To deliver the exertion in the virtual environment, the paddle rotations drive the dragon boat's movements, replicating the dragon

boat in reality. As such, our system contains three paddling techniques that emulate the paddles' rotations and, hence, the force driving the dragon boat's directional movements. The dragon boat sprints or reverses when both paddles have the same force level. Otherwise, it spins leaning left or right, depending on the dominant force among the paddles. Figure 3 shows the user's first-person view and the user interfaces depicting the navigation signifier (in meters) and paddle angles.

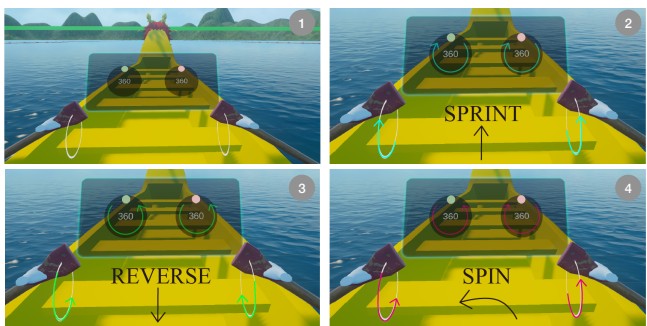

**Figure 3: 3D UIs for (1) paddling controls and boat navigation, where the two circles depict the paddling angles, e.g., 337°, controlling the boat movements, e.g., (2) Sprint, (3) Reverse and (4) Spin.**

These techniques, representing various physical fitness demands, aim to offer the dragon boat culture to a wider audience of diversified fitness levels. Participants, depending on their fitness level, can rotate the dragon boat paddles through their thumbs, forearms, or full arms (with small movements of their upper bodies). In addition, to investigate the performance of our proposed controllers (Exertion Controller) designated for the virtual dragon boat, two widely used interaction techniques (Joystick Controller and

IMU Controller), enabled by the commercial controllers of Oculus Quest, were chosen as the baseline, with the following details.

**Joystick Controller** (JC): This technique requires the user's thumbs to work subtly on the joysticks on the controllers that guide the directional movements of the virtual dragon boat while the user, perhaps with rested arms and a sitting posture. The upward and downward movements of the joystick lead to the paddle's anticlockwise and clockwise rotations, respectively. In other words, the paddle of anticlockwise rotation can drive the dragon boat forward, or vice versa (Figure 4). The angular velocity of the dragon boat maps directly to the joystick magnitude, e.g., pushing the joystick to the very end gives the maximum angular velocity.

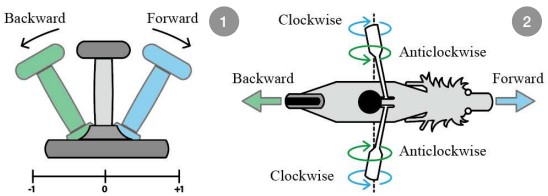

**Figure 4: Joysticks (1) for paddling and movements (2).**

**IMU Controller** (IC): This paddling technique introduces a greater extent of body movement than JC. Users with IC have to move their forearms constantly to rotate the paddles in a sedentary posture, and the inertial measurement unit (IMU) inside the controller detects the changes in forearm positions. Such positional changes map to the angular movements of the paddles, as shown in Figure 5. Thus, the clockwise rotation of the controllers directs the dragon boat to move backwards, while the anti-clockwise rotation of the controllers leads to the dragon boat's forward directional movements. Moreover, as rotating paddles drive the dragon boat's movement speed, the more the controllers rotate, the higher the velocity of the dragon boat.

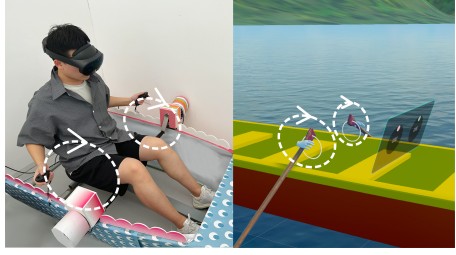

**Figure 5: Forearm mid-air movements with the standard controllers of Oculus Quest Pro.**

**Exertion Controller** (EC): The wheels attached to a DC-geared motor connect to a tangible handle that allows a user to make rotational movements mimicking the paddle. When the user rotates the wheels attached to the EC, an absolute encoder of the system measures the user's rotational data, which is mapped onto the rotation of a paddle within the virtual environment. An anti-clockwise rotation of the paddle propels the boat forward, while a clockwise rotation causes the boat to move backwards. Upon the

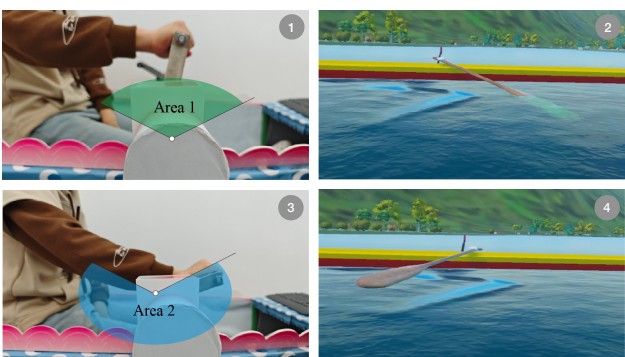

**Figure 6: The handles (EC) controlling virtual paddles.**

paddle hitting the water in the virtual environment, users experience resistance as feedback, which is generated by the DC-geared motor, mimicking the rowing experience in the real world. Thus, users with the technique move their upper body and full arms to drive the boat movements. To accelerate the dragon boat in the virtual world, users can rotate the wheels more rapidly, consequently increasing physical exertion. In other words, the more rapid the rotation of the EC wheels, the higher the dragon boat velocity.

### 3.3 Configurations and Hardware Controller

Regarding the *Exertion Controller* (EC) described in the previous paragraph, we design and build a hardware prototype of paired rotational handles (Figure 2 (1)) that aims to replicate the experience of dragon boat rowing. The prototype comprises two metal spinning handles controlled by an Arduino-based microcontroller unit (MCU). A pivotal component of the prototype is the integration of an absolute encoder with the wheel mechanics that connect to the rotating handles. This encoder is instrumental in tracking the paddle's angular position within the VR environment, ensuring a high degree of realism in the metaverse. Figure 6 depicts a user with EC in a sitting posture who rotates the handles continuously to engage with the paddles of the virtual dragon boat.

The absolute encoder attached to the handles keeps track of the handle's angular changes. Meanwhile, such changes can be translated into corresponding angular motions of the virtual paddles. Accordingly, the MCU captures signals from the encoder and transmits them to a relay system that provides resistance as haptic feedback simulating the virtual paddles thrashing in the water. Figure 6 (1 & 2) shows that the handle within Area 1 ($290° - 360°$ & $0° - 70°$) activates the emulation of water resistance encountered by the virtual paddles. In contrast, Figure 6 (3 & 4) indicates the handles within Area 2 ($70° - 290°$) deactivates the water resistance as the virtual paddles hover mid-air.

### 4 EVALUATION OF PADDLING TECHNIQUES

Our study focuses on understanding the user experience and exertion of the three paddling techniques in the digital twin campus and investigating their comparative advantage. We briefly highlight the key difference between the paddling techniques, as follows. First, the joystick condition required the least physical activity among

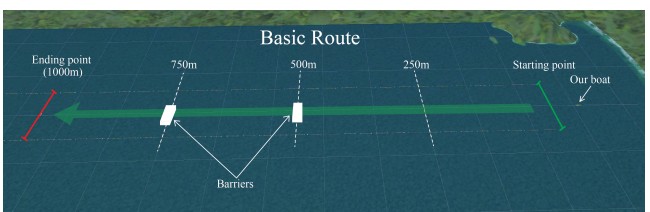

**Figure 7: 1km-rowing task and two barriers (500m & 750m).**

the techniques because of sitting posture and thumb movements. Second, the users with IC would sit on the chair integrated into the system. The user's forearm then rotates the controllers without constant resistance, i.e., like passive haptic feedback [44]. They have a degree of freedom similar to rowing the boat's oars. In contrast, EC integrates precise motion detection and tactile resistance feedback, aiming to mimic the real-world rowing experience. Thus, the user with EC would rotate the wheel of our dragon boat system to overcome the resistance during the boat riding. Therefore, the main difference between IC and EC is the availability of haptic feedback and resistance. In summary, these three paddling techniques, spanning from basic to advanced simulation fidelity, allow us to analyse how different designs of interactive multimedia affect the perceived workload, sickness and user experience of the VR environment. By comparing these techniques, the research aims to gain insights into the physical demands of a wide range of amateur users and offer design considerations for the virtual experience of dragon boats.

### 4.1 Study Design

The study was conducted using a within-subject design with three independent variables. As independent variables, the paddling techniques as above are JC, IC, and EC. They were *user experience* measured with the 8-item User Experience Questionaire Short Form (UEQ-S) with a 1-7 Likert scale [50], simulator sickness from the 16-item Simulator Sickness Questionnaire (SSQ) with a 0-3 Likert scale [26], and workload measured with the standard NASA Task Load Index (TLX) with a 1-7 Likert scale [21]. In addition to these questionnaires, the Average Heart Rate (HR) data was measured by a Smart Fitness Band. Due to age differences, individuals have different maximum heart rates, so we normalize the heart rate from 0 to 1. For a paddling technique, this value represents the collected heart rate among individuals. It is computed by an individual's maximum heart rate (varied by one's age), equivalent to $avgHR = 211 - (0.64 \times$ an individual's age) [36]. Then, we sum up the participants' heart rate and calculate the normalized average.

On the other hand, we conducted a semi-structured interview consisting of 4 topics: (1) Participants' preferences and overall assessment of these systems; (2) Expectations regarding users' knowledge about the virtual dragon boat and how they will use it; (3) Suggestions for system improvements; (4) Transferability of learning to real-world rowing. For each topic, we include a fixed open-ended question followed by several questions based on the previous answers from the participants. The interviews were recorded. Finally, the records were transcribed into transcripts.

### 4.2 Procedures

The study occurred in a university lab prepared to resemble a living room scenario containing a dragon boat, a monitor, and the Meta Quest Pro. Before beginning the study, each participant signed an informed consent form. After a brief introduction, participants completed a demographic form providing basic information.

Initially, the procedure starts with a calibration phase that allows users to adjust to the virtual dragon boat. Before beginning the virtual experience, participants press a button on the controller to reset the view in the head-mounted display (HMD). Subsequently, they press the trigger on the controller to reset their position within the virtual boat, ensuring optimal exertion during the rowing motion on the dragon boat's wheel. Following the calibration phase, the game proceeds to a training phase, during which participants can practice the rowing skills acquired from video tutorials for approximately 10 minutes. Once the participants have become sufficiently familiar with the rowing techniques in each condition, they can begin the formal rowing race sessions. They played all three possible permutations of our independent variables (JC × IC × EC). It should be emphasised that every participant attended a three-day session, during which one paddling technique was assessed per day. The order was counterbalanced using the Latin Square [51]. After each session, participants completed the questionnaires (Section 4.1) measuring their experience in the virtual environment. The three-day session took, on average, 1 hour. After finishing the three techniques on Day 3, We concluded three paddling techniques by the semi-structured interview. After the interview, each participant received cash (7 USD) and a gift card (14 USD). The experiment was authorised by the University Ethics Committee.

### 4.3 Task

Participants have been told to complete the dragon boat trail as quickly as possible. Their performance is measured quantitatively by the completion time. No repetition is required for each paddling technique. Figure 7 depicts the water trail. The first half of the trail (0 – 500 metres) is a straight line, which aims to evaluate their speed and navigational efficiency, primarily with a spinning dragon boat. Moreover, the second half (500 – 1000 metres) has two barriers that induce the participants to circumvent the wall-shaped obstacles. Thus, rowing skills such as the dragon boat's spinning and reversing become necessary. This assesses the participant's ability to maintain an appropriate balance between speed and the rowing skills acquired through VR exertion, testing their capacity to manage competing demands within the virtual environment.

### 4.4 Participants

We recruited 18 participants (Gender: 8F, 10M & Age: $Mean = 24.83, SD = 2.995$) from a local university through posters and social media. Six of them rated themselves as experienced users of VR systems, and the others (12p) reported little to no experience. We used the 18-item Motion Sickness Susceptibility Questionnaire Short-form (MSSQ-Short) [16] to measure participants, and no participants reported elevated susceptibility for motion sickness with an average score of 11.56 ($SD = 11.36$), which shows that no significance exists.

# 5 RESULTS

*Average Heart Rate Percentage and Calories*. A one-way RM-ANOVA revealed significant effects of the experimental condition on both Average Heart Rate Percentage (Avg HR%) ($F(2, 34) = 46.9756, p < 0.0001, \eta^2 = 0.734$) and Calories burned ($F(2, 34) = 42.1643, p < 0.0001, \eta^2 = 0.713$). Figure 8 shows that both IC and EC resulted in similar heart rates, but JC had a significantly lower heart rate than them. Post-hoc analysis with Bonferroni corrections for Avg HR% indicated no significant difference between EC and IC conditions ($p = 0.8007$). Still, significant differences were observed between the JC and both EC ($p < 0.0001$) and IC ($p < 0.0001$) conditions, with the JC condition leading to a significantly lower Avg HR%. These results suggest that the physical activities simulated by the IC and EC conditions are more demanding than the JC condition, as evidenced by the higher Avg HR%. Similarly, post-hoc analysis for Calories burned, adjusted for the non-normal distribution through the Aligned Rank Transform (ART) [63], revealed significant differences between all conditions ($p < 0.0001$) for JC vs. IC and JC vs. EC), and ($p = 0.0227$) for IC vs. EC), with the IC and EC conditions leading to higher energy expenditure than the JC condition. The IC condition was also found to result in significantly higher Calories burned compared to the EC condition, albeit to a lesser extent. Our findings imply increased heart rates and energy expenditure by IC and EC, as both features with arm movements. Interestingly, the mid-air rotating gestures of IC lead to higher calories burnt over EC (Figure 8), as we observe that the users meticulously utilize the leverage from the hardware device attached to the dragon boat's basement to conserve their efforts.

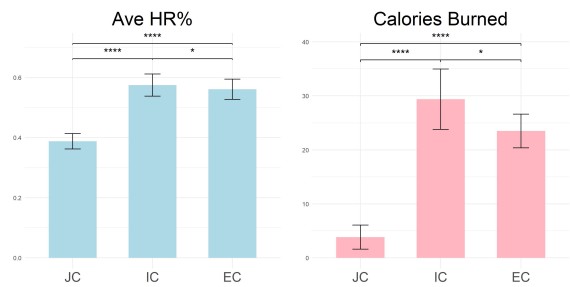

**Figure 8: Comparison of Average Heart Rate Percentage and Calories burned across three conditions: JC, IC, and EC. Bars represent means with error bars depicting 95% confidence intervals. Significant differences between conditions are annotated above the bars, with '*' and '****' indicating significance levels at $p < 0.05$ and $p < 0.0001$, respectively.**

*User Experience Questionnaire*. A Friedman test was first applied to assess the differences across conditions for each metric. The results indicated a significant effect of the condition on Pragmatic Quality ($\chi^2(2) = 6.426, p = 0.040$), highlighting variability in how each condition supported the users' practical goals. However, no significant differences were found for Hedonic Quality ($\chi^2(2) = 5.460, p = 0.065$) and Overall Satisfaction ($\chi^2(2) = 3.686, p = 0.158$), suggesting similar levels of enjoyment and overall user satisfaction across conditions. Figure 9 presents these findings

on a scale from 0 (Worst) to 3 (Best). Given the statistical significance of Pragmatic Quality, post-hoc analysis was conducted using the Wilcoxon Rank-Sum Test (Mann-Whitney U test) with Bonferroni correction for multiple comparisons. The post-hoc comparisons yielded a significant difference in Pragmatic Quality between the JC and IC conditions ($U = 243.5, p = 0.0096$), with a Bonferroni adjusted alpha level of 0.0167, indicating that JC significantly better supports practical user goals than IC. However, there is no significant difference in Pragmatic Quality between JC and EC ($U = 212.5, p = 0.1075$) and between IC and EC ($U = 124.5, p = 0.2390$), indicating that the support for practical user goals did not significantly vary between these conditions.

These results imply that the nuanced impact of hedonic quality exists among conditions, i.e., fun and joy. Nonetheless, the pragmatic quality shows preliminary evidence that JC's utility and usability outperformed its counterpart, IC. Users show a preference for physical engagement in VR rowing, albeit with moderated exertion intensity to avoid undue fatigue. This indicates a preference for physical engagement in VR rowing within limits that balance immersion and comfort, avoiding excessive strain to maintain user engagement without leading to fatigue.

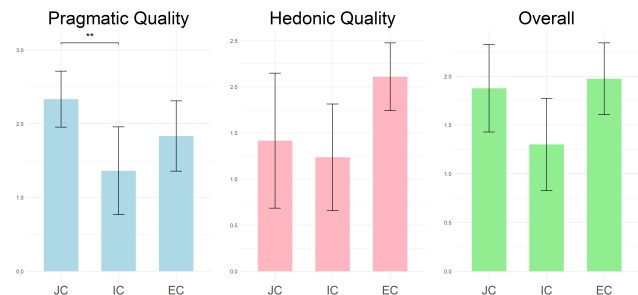

**Figure 9: Comparison of Pragmatic Quality, Hedonic Quality, and Overall Satisfaction scores across three conditions: JC, IC, and EC. Each bar represents the mean score, with error bars showing 95% confidence intervals. '**' denotes a significant difference between conditions for Pragmatic Quality ($p < 0.01$) following the Friedman test and post-hoc Wilcoxon Rank-Sum Test with Bonferroni correction.**

*NASA Task Load Index*. A Friedman test assessed the statistical significance of differences across conditions for each NASA-TLX metric. The results (Table 1) indicated significant differences in Mental Demand ($\chi^2(2) = 6.70, p = 0.035$), Physical Demand ($\chi^2(2) = 14.69, p < 0.001$), Temporal Demand ($\chi^2(2) = 11.83, p = 0.003$), Effort ($\chi^2(2) = 12.00, p = 0.003$), and Frustration ($\chi^2(2) = 8.76, p = 0.013$), with no significant difference observed in Performance ($\chi^2(2) = 0.047, p = 0.977$).

Subsequently, a post-hoc analysis employing the Wilcoxon Rank-Sum Test (Mann-Whitney U test) with Bonferroni correction revealed significant differences between conditions for several metrics except for Performance (due to the absence of statistical significance). Specifically, as shown in Table 2, Physical Demand and Effort exhibited significant differences when comparing JC with both IC ($U_{Physical} = 82.0, p = 0.009; U_{Effort} = 72.5, p = 0.004$)

**Table 1: Friedman Test Results for NASA-TLX Metrics Analysis, incl. Mean (SD), with '\*', '\*\*' and '\*\*\*' indicating significance levels at $p < 0.05$, $p < 0.01$ and $p < 0.001$, respectively.**

| Metric | JC | IC | EC | $\chi^2(2)$ | $p$-value |
|---|---|---|---|---|---|
| Mental Demand | 2.11 (1.37) | 3.06 (1.73) | 2.72 (1.74) | 6.70 | 0.035 (*) |
| Physical Demand | 1.94 (1.63) | 3.61 (2.00) | 4.22 (1.63) | 14.69 | <0.001 (***) |
| Temporal Demand | 2.06 (1.26) | 2.78 (1.63) | 3.44 (1.72) | 11.83 | 0.003 (**) |
| Performance | 3.00 (2.14) | 3.06 (1.95) | 2.67 (2.03) | 0.047 | 0.977 |
| Effort | 2.28 (1.84) | 3.94 (1.35) | 4.44 (1.69) | 12.00 | 0.003 (**) |
| Frustration | 1.39 (0.85) | 2.28 (1.60) | 2.28 (1.81) | 8.76 | 0.013 (*) |

**Table 2: Post-Hoc Analysis Results for NASA-TLX Metrics. '\*', '\*\*', and '\*\*\*' denote significance levels at $p < 0.05$, $p < 0.01$, and $p < 0.001$, respectively, following the Wilcoxon Rank-Sum Test with Bonferroni correction.**

| NASA-TLX Metric | Comparison | $U$-value | $p$-value |
|---|---|---|---|
| Mental Demand | JC vs IC | 107.5 | 0.234 |
| | JC vs EC | 132.0 | 0.993 |
| | IC vs EC | 182.0 | 1.586 |
| Physical Demand | JC vs IC | 82.0 | 0.009 (**) |
| | JC vs EC | 53.5 | <0.001 (***) |
| | IC vs EC | 133.0 | 0.359 |
| Temporal Demand | JC vs IC | 118.0 | 0.154 |
| | JC vs EC | 87.5 | 0.016 (*) |
| | IC vs EC | 125.0 | 0.241 |
| Effort | JC vs IC | 72.5 | 0.004 (**) |
| | JC vs EC | 60.0 | <0.001 (***) |
| | IC vs EC | 127.5 | 0.270 |
| Frustration | JC vs IC | 110.5 | 0.180 |
| | JC vs EC | 119.5 | 0.338 |
| | IC vs EC | 167.5 | 2.591 |

and EC ($U_{Physical} = 53.5, p < 0.001; U_{Effort} = 60.0, p < 0.001$). Additionally, Temporal Demand was significantly higher for EC than JC ($U_{Temporal} = 87.5, p = 0.016$). The results imply that IC and EC successfully introduced dragon boat users to appropriate and purposeful exertion regarding physical demands and efforts. Additionally, such exertions are at the right balance as all methods share similar temporal and mental demands as well as frustration.

***Simulator Sickness Questionnaire.*** A Friedman test was applied to assess the statistical significance of differences across conditions for each Simulator Sickness Questionnaire (SSQ) metric, including Nausea (N), Oculomotor (O), Disorientation (D), and Total Score (TS). Table 3 lists the mean value and standard deviation of such values ranging from 0 (best) to 48 (worst), demonstrating reasonably low levels of sickness among all conditions. For the Nausea subscore, the Friedman test indicated no significant score difference across the conditions ($\chi^2(2) = 1.75, p = 0.417$). Similarly, no significant differences were found for the Oculomotor subscore ($\chi^2(2) = 0.39, p = 0.823$), the Disorientation subscore ($\chi^2(2) = 1.77, p = 0.412$), or the Total Score ($\chi^2(2) = 0.56, p = 0.755$). The results imply that participants did not experience significant differences in simulator sickness symptoms, including nausea, oculomotor issues, or disorientation across all conditions. In other words, increased exertion with forearm movements does not evoke

additional simulator sickness in virtual dragon boat racing , thus delivering acceptable and low-nausea experiences.

**Table 3: Friedman Test Results for Simulator Sickness Questionnaire (SSQ) Metrics, incl. Mean (SD).**

| Metric | JC | IC | EC | $\chi^2(2)$ | $p$-value |
|---|---|---|---|---|---|
| Nausea | 14.84 (24.39) | 23.32 (42.06) | 16.96 (24.75) | 2.326 | 0.3127 |
| Oculomotor | 15.58 (17.49) | 16.00 (26.63) | 13.90 (21.04) | 1.541 | 0.4631 |
| Disorientation | 17.01 (20.95) | 18.56 (33.08) | 11.60 (23.66) | 4.769 | 0.0924 |
| Total Score | 177.40 (213.96) | 216.48 (365.92) | 158.79 (244.52) | 4.308 | 0.1161 |

***Completion Time.*** A Friedman test was applied to assess the statistical significance of differences across conditions for completion time (second, s) with a Chi-square test indicating statistical significance exists $\chi^2(2) = 21.78, p < 0.001$. Specifically, participants with JC ($Mean = 197.72s, SD = 20.57s$) completed the task faster than those with IC ($Mean = 335.64s, SD = 109.85s$) and EC ($Mean = 282.29s, SD = 46.78s$) conditions. The post-hoc analysis employing the Wilcoxon Rank-Sum Test (Mann-Whitney U test) with Bonferroni correction revealed that statistical difference exists between EC and JC ($p < 0.001$), and the IC and JC ($p < 0.001$). Conversely, no statistical significance was found between EC and IC ($p = 0.319$). The results suggest a comparable performance between two similar arm-based paddling techniques, aligning with the Performance metric (w/o stat. diff.) based on the NASA-TLX questionnaire (Table 1).

***After-test Interview.*** The thematic analysis using user interview data via Nvivo [62] revealed several significant themes pertinent to evaluating the VR dragon boating experience. These themes encompassed the following three themes.

(1) Theme 1: Immersion and realism of the virtual experience: The immersion and realism of the virtual experience was a prominent theme among participants. Participants reported that the VR environment and sound effects contributed to a heightened sense of immersion and realism. [P2], for instance, remarked, *"First of all, the perspective of this VR looks quite realistic, giving a bit of a sense of being there. Then with the sound effects, the overall feeling is that the sense of participation is still quite strong."* However, it is important to note that some participants, such as [P2, P6], also observed that the artificial nature of the scene was still noticeable, which could potentially detract users from the immersive experience. This suggests that further improvements in the graphics of the virtual environment may be necessary to fully engage users and provide a truly immersive experience.

(2) Theme 2: Physical challenges and discomfort: Several participants discussed the physical demands and discomfort associated with the MetaDragonBoat system, particularly in relation to motion sickness and fatigue. [P2] stated, *"I personally get vertigo in virtual worlds very easily, and it can be quite difficult for me to continue paddling forward in this somewhat dizzy state."* Similarly, [P17] found the paddling technique named EC to be *"tiring than what I expected."* These findings indicate that the MetaDragonBoat system may need to be adjusted to accommodate users who are more susceptible to motion sickness or fatigue. Alleviating these physical challenges could potentially improve the user acceptance

of the technology and make the virtual experience more accessible to a wider range of individuals.

(3) THEME 3: TRANSFERABILITY OF SKILLS TO REAL-WORLD ROW-ING: While some participants, such as [P1, P15], believed that the knowledge gained could be transferable, others, like [P6, P18], were less certain due to the differences between the virtual and real-world rowing conditions. [P6] noted, *"Basically, I think there is room for learning the rowing skills with EC; I've actually seen it on TV, but normally, if you want to row a dragon boat, a professional device in virtual reality is a starting point of demonstrating the professional dragon boat racing, like how to row. I understand that the current device limits to just using hands to row; the next iteration should consider the movement of other large muscles like the real dragon boat rowing."* This suggests that further refinements may be necessary to better align the virtual experience with real-world rowing conditions and facilitate skills transfer. Addressing these concerns could make a more effective system that encourages the participation of dragon boat activities in the real world.

## 6 CONCLUSION AND DISCUSSION

This paper implemented a digital replica of a university campus and dragon boats, an integral component of China's cultural heritage. Within the context of a digital twin, we have created a virtual experience called Meta Dragon Boat. As dragon boats have historical associations with demanding and physically challenging tasks, we created a specialised hardware device specifically for dragon boat paddling, and conducted a comprehensive assessment of three different paddling approaches. Our study's results offer clues to achieving an equilibrium between user satisfaction and physical effort in a virtual experience. We discuss our main findings, limitations, and future work in the following paragraphs.

***Paddling techniques and target users***. One key purpose of MetaDragonBoat is to promote cultural heritage to audiences beyond the elite sportsmen. Our experiment measured the heart rate of 18 healthy adults, and the user's heart rate with the three paddling techniques ranged from 60 to 139. The mean and standard deviation values (beats/min) of the three paddling techniques are as follows: JC (Mean: 81.63, SD: 12.38); IC (Mean: 110.78, SD: 13.05); and, EC (Mean: 106.51, SD: 13.55). Our proposed paddling techniques introduce appropriate exertion, supported by Ostchega et al.'s work [39]. Their work found that the mean resting pulse (beats/min) for children, adolescents, adults and elderly are 96, 78, 72, and 70, respectively, while the maximum resting pulse (beats/min) for children, adolescents, adults and elderly are 114, 94, 92, and 90, respectively.

The three distinct paddling techniques allow individuals of various physical fitness to participate virtually in the cultural journey of dragon boat racing. In other words, users can choose their paddling techniques accordingly. Among the three paddling techniques, the first technique (JC) is regarded as an easy technique that requires only subtle thumb movements exerted on the joysticks of the standard controller while the user is situated in a relaxed posture. As such, this method is primarily oriented to inspire participation and encourage those being physically compromised, e.g., the elderly and children. The other paddling techniques (IC and EC) encourage a more rigorous physical endeavour, leading to elevated average heart rates. Therefore, such techniques have the potential to benefit

the majority of adolescents and adults who have a satisfactory level of physical fitness.

***Experience or Exertion?*** The recent development of metaverse applications primarily focuses on bringing positive experiences, convenience and utility, including senses of presence and realism, user's wellness, virtual community as well as remote collaboration [12, 60, 61]. The common practice of designing a metaverse system is reducing user workload and effort. On the contrary, we have a counter-intuitive approach of imparting extra workload and efforts to the users' virtual experience, due to the nature of the dragon boat culture in the Lingnan region [13]. Dragon boat culture exemplifies the fusion of deity worship with physical activity, similar to how Western mythology and other ancient religions also value sporting activities [17], such as the gods of athletics known as Nike and Herme [42]. More importantly, our work attempts to pave a path towards balancing user experience and exertion among many future metaverse applications. Our study, therefore, considers metrics relevant to user experience and exertion. While it is not possible to definitively conclude that user experience and exertion will not conflict with each other in other metaverse applications, our evaluation of MetaDragonBoat in Section 5 demonstrates the possibility of maintaining usability and usefulness even when physical demands and efforts are elevated appropriately. Furthermore, we have seen no discernible negative effects on the user's perception and their level of performance.

***Limitations***. We have identified several limitations in our current study, as follows. First, apart from the small participant size, we recognise that the results are limited to young adults due to campus recruitment and social networks, while middle-aged and older adults, as well as the elderly, are neglected. Incorporating a diverse age range of individuals may assist us in maintaining the generalizability of our findings. Second, the resistance encountered during race-like training in dragon boating directly correlates with the level of effort and, thus, user fatigue. Meanwhile, motion sickness in VR could cause nausea and discomfort. Nonetheless, an in-depth understanding of these factors requires assessing extended periods of virtual experience, e.g., recording user data throughout 15-day training sessions. Last, we acknowledge that the racing element in our experiment is only a single facet of the entire dragon boat culture. Nonetheless, the impacts of other elements need further investigation. It is worthwhile to highlight that this aquatic team activity fosters connections among individuals, leading to the establishment of a virtual community. This community offers a range of social activities, including virtual gatherings, socialising, and co-ordinating dragon boat excursions inside the virtual campus. These societal elements may influence the user perception of such virtual environments.

***Future Work***. First, we plan to conduct an extended study to gain a comprehensive understanding of the user dynamics inside a multi-user environment. We will also re-run the study with individuals from other age groups, especially older adults, recognising that the current work's participants are in their early twenties. Third, the studied paddling techniques and their variations will be further examined for realism and exertion as an enjoyable and adrenaline-pumping water sports endeavour.

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
