# OpenReview forum: "MetaDragonBoat: Exploring Paddling Techniques of Virtual Dragon Boating in a Metaverse Campus"
_acmmm.org/ACMMM/2024/Conference — MM2024 Poster_

### Official Review · Reviewer_yV5P · 2024-05-21

**Rating:** 2
**Confidence:** 4

**Summary:**

The paper reports MetaDragonBoat - a VR based system which simulates, through haptic feedback in VR, traditional Chinese Dragon Boat racing. The application is evaluated with a sample of 18 users with encouraging results.

**Strengths:**

SDGs are important and MetaDragonBoat is one initiative in this sense.

**Limitations:**

As a paper submitted to the research track of a prestigious and highly selective conference such as ACM MM, one expects there to be an underpinning (and novel) research question. This is missing in the paper. Designing (and evaluating) a VR application with novel features is not in itself proof of research novelty.

Moreover, there is an existing body of work on rowing in VR (mostly with haptic feedback) which the authors seem to be unaware of:

1. Li, Xuecheng, Zhengyu Wu, and Ting Han. "Gamification-Based VR Rowing Simulation System." Human-Computer Interaction. Recognition and Interaction Technologies: Thematic Area, HCI 2019, Held as Part of the 21st HCI International Conference, HCII 2019, Orlando, FL, USA, July 26–31, 2019, Proceedings, Part II 21. Springer International Publishing, 2019.
2, Parton, B. J., & Neumann, D. L. (2019). The effects of competitiveness and challenge level on virtual reality rowing performance. Psychology of Sport and Exercise, 41, 191-199.
3. Shoib, N. A., Sunar, M. S., Nor, N. N. M., Azman, A., Jamaludin, M. N., & Latip, H. F. M. (2020). Rowing Simulation using Rower Machine in Virtual Reality. In 2020 6th International Conference on Interactive Digital Media (ICIDM) (pp. 1-6). IEEE.
4. Van Delden, R., Bergsma, S., Vogel, K., Postma, D., Klaassen, R., & Reidsma, D. (2020). VR4VRT: virtual reality for virtual rowing training. In Extended Abstracts of the 2020 Annual Symposium on Computer-Human Interaction in Play (pp. 388-392).

Another very recent paper which might be of interest to the authors is:
Hedlund, M., Bogdan, C., Meixner, G., & Matviienko, A. (2024). Rowing Beyond: Investigating Steering Methods for Rowing-based Locomotion in Virtual Environments. In Proceedings of the CHI Conference on Human Factors in Computing Systems (pp. 1-17).

Even if one ignores the above papers not being referenced, there is one research angle that could have been adopted by the authors - they make the claim that MetaDragonBoat was specifically designed for rowing in Dragon Boat racing. Although it is unclear from the paper what exact rowing characteristics specific to Dragon Boat racing were included in MetaDragonBoat, one way that the authors could have included a research angle is to have two versions of the application, one with rowing/haptic feedback specifically tailored to Dragon Boat racing, and another with normal/control rowing/haptic feedback (perhaps as described in one of the above papers) and the research question might ask if tailoring this rowing haptic feedback does result in a better user experience.

A last comment - the authors seem to use VR and Metaverse interchangeably and synonymously. They are not identical, or, if the authors consider this to be the case, they should explain why.

**Suitability:**

2

---

### Official Review · Reviewer_ZBj7 · 2024-05-21

**Rating:** 5
**Confidence:** 3

**Summary:**

The contributions of this work are at the intersection of immersive technologies and cultural heritage. First, the study developed a physical mock-up of a dragon boat with aesthetic designs that accurately mirrors a virtual dragon boat within a digital twin of a university campus. This integration allows users to experience the cultural heritage of dragon boat racing from an indoor environment, effectively combining physical and virtual experiences. Second, the project introduced hardware interaction techniques supported by haptic feedback. These techniques include physical paddle controllers that simulate the physical actions and water resistance of dragon boat rowing, balancing user experience and physical exertion. Third, a comprehensive user evaluation involving 18 participants revealed that the physical paddle controllers significantly increased heart rates and workloads compared to standard commercial controllers. Despite the increased physical demands, the new hardware did not negatively impact motion sickness, maintaining usability and utility comparable to traditional VR methods. This evaluation highlights the system's effectiveness in delivering an immersive and physically engaging cultural experience, fulfilling the objectives of preserving cultural heritage and fostering spiritual dedication.

**Strengths:**

Well-Designed System Implementation: The reasoning and motivation behind the implementation of the MetaDragonBoat system are robust and thoughtfully conceived. The system aims to address cultural preservation by making the traditional sport of dragon boat racing accessible in a virtual format, which aligns well with the goal of safeguarding intangible cultural heritage. The implementation reflects a deep understanding of both the cultural significance of dragon boating and the potential of immersive technologies to create engaging and educational experiences.

Comprehensive Applications and System Evaluation: The applications and system evaluation processes are meticulously planned and executed. The study evaluates the effectiveness of different paddling techniques and the user experience in a virtual environment, using both commercial controllers and custom-designed physical paddle controllers with haptic feedback. This thorough approach ensures that the system not only simulates the physical aspects of dragon boat racing accurately but also provides insights into user preferences and the physical demands of different techniques. The inclusion of various control methods allows for a comprehensive assessment of the system's adaptability and user engagement.

Solid Data Analysis: The data analysis component of the study is rigorous and well-structured, providing robust evidence to support the findings. The analysis includes detailed measurements of heart rates and user feedback, offering a clear picture of the physical exertion and user satisfaction associated with each paddling technique. The use of quantitative data, combined with qualitative user feedback, strengthens the validity of the results and helps identify areas for improvement. This solid analytical foundation ensures that the conclusions drawn from the study are reliable and informative, guiding future enhancements of the MetaDragonBoat system.

**Limitations:**

Participant Demographics: The study's findings are limited by the small participant size and the homogeneity of the participants, who were primarily young adults recruited through campus networks. This excludes middle-aged, older adults, and the elderly, affecting the generalizability of the results.

Effort and Fatigue: The resistance encountered during virtual dragon boat racing correlates with user effort and fatigue. Additionally, motion sickness in VR can cause nausea and discomfort. A thorough understanding of these issues requires longer study periods, such as 15-day training sessions.

Cultural Representation: The experiment focused solely on the racing aspect of dragon boat culture, neglecting other cultural elements. The virtual community aspect, which includes social activities and team coordination, also plays a significant role and needs further exploration to understand its impact on user perception.

**Suitability:**

2

---

### Official Review · Reviewer_eeU5 · 2024-05-24

**Rating:** 2
**Confidence:** 3

**Summary:**

This paper introduces a Cultural Heritage VR exergaming experience. The installation aims at replicating a traditional boat race and, through experimentation of different controllers in their level of exertion and haptic feedback, offer different insights into the cultural experience. It describes the physical exergaming interface as well as user experiments investigating physiological impact, cognitive load, objective performance and user experience.

**Strengths:**

Complex fully-implemented installation blending content elements (cultural heritage) with human-centric approach.
Generally well-designed experiment, with good level of statistical skills in the results' analysis.
Originality of the exergaming concept in terms of Cultural Heritage.

**Limitations:**

On relevance and suitability: while VR and the cultural element could be seen as relevant to MM, focus on user experience and exergaming is, in my view, not. On the grounds that the central element of media, namely -consumption- does not seem compatible with this mode of interaction. The investigation of the experimental conditions through controllers would make this paper more relevant to ACM VRST or possibly IUI.

Overall, findings are consistent with previous literature, but the contribution is moderately significant, confirming the relation between device and exertion but also the 'flow-like' balance between effort and enjoyment which is typical of VR and gaming installations and may differentiate them from simulation and training.

The investigation of cybersickness could be more rigorous, especially since it transpires in one of the subjects' narrative comments. Firstly, a complete SSQ score could be better analyzed. Secondly, potential impact of SS on HR measurements should have been discussed in view of parasympathetic activation.

It is peculiar that the paper has not resorted to a better measure of presence or immersion from the perspective of exploring quantitatively the impact of exertion on presence. This is of particular relevance since the findings tend towards a compromise between enjoyment and exertion.

The list of limitations reported by the user themselves is rather impressive, and would have suggested perhaps a different focus for the paper.

**Suitability:**

1

---

### Official Review · Reviewer_3QWC · 2024-05-25

**Rating:** 6
**Confidence:** 3

**Summary:**

The paper emphasizes preserving cultural heritage for sustainable urban development per the UN's SDGs. It proposes using Virtual Reality (VR) to enhance the Dragon Boat Festival's preservation and accessibility. The MetaDragonBoat solution allows virtual participation in dragon boat racing, creating an immersive experience with haptic feedback paddle controllers. Three paddling techniques for boat movement in the virtual world for diversified users were tested and evaluated.
This approach enables realistic rowing simulations, promoting broader appreciation of this cultural tradition.

**Strengths:**

- The idea of integrating resistance into the paddle controls allows users to simulate the physical exertion of dragon boat racing. This provides a deeper understanding and appreciation of this cultural heritage.
- A very careful and informative evaluation of the three different controllers

**Limitations:**

- In the Dragon Boat Festival, paddlers use the "single-side paddling" technique. This involves each paddler using a single-bladed paddle to paddle exclusively on one side of the boat. This coordinated effort on either side of the boat helps maintain the boat's speed and direction. In the proposed digital twin, two oars are used, which makes the preservation of the Dragon Boat tradition less faithful. Are you going to address this issue?

-- minor: References
- ref. 7: add pages number and doi: 1146–1155. https://doi.org/10.1145/2964284.2964293
ref. 11: complete  In: Nishita, T., Peng, Q., Seidel, HP. (eds) Advances in Computer Graphics. CGI 2006. Lecture Notes in Computer Science, vol 4035. Springer, Berlin, Heidelberg. https://doi.org/10.1007/11784203_25
- ref. 11: do not use all the initial capital letters
- ref. 22: article n. 49,1-2. https://doi.org/10.1145/3607822.3618022
- ref. 25: Kennedy, R. S., Lane, N. E., Berbaum, K. S., & Lilienthal, M. G. (1993). Simulator Sickness Questionnaire: An Enhanced Method for Quantifying Simulator Sickness. The International Journal of Aviation Psychology, 3(3), 203–220. https://doi.org/10.1207/s15327108ijap0303_3
- ref. 32: Ph.D. Dissertation. The George Washington University.
- ref. 33: “sigchi” => “SIGCHI”
- ref. 36: “F itness S tudy” => “Fitness Study”
- ref. 39: Natl Health Stat Report. 2011 Aug 24;(41):1-16
- ref. 42:  => In Paul and the agon motif: traditional athletic imagery in the Pauline literature, 16–22, https://doi.org/10.1163/9789004265936_003
- ref. 48: => Pages 123–130. https://doi.org/10.1145/199404.199426
- ref. 54: => INSA de Rennes
- ref. 66: 116 => 116-125. https://doi.org/10.9734/bpi/mono/978-81-19039-58-6/CH10
- ref. 66: “Traditions and Cultural Heritage: Genesis, Reproduction, and Preservation (2023), 116.“ => “In Traditions and Cultural Heritage: Genesis, Reproduction, and Preservation (2023), 116-125. https://doi.org/10.9734/bpi/mono/978-81-19039-58-6/CH10

**Suitability:**

3

---

### Meta-Review · Area_Chair_SSGN · 2024-07-02

**Recommendation:** Accept (Poster)
**Confidence:** 4

**Metareview:**

The paper presents a VR-based system, MetaDragonBoat, aimed at preserving the cultural heritage of the Dragon Boat Festival by simulating traditional dragon boat racing in a virtual environment. The system incorporates haptic feedback through paddle controllers, offering users an immersive and physically engaging experience. The study comprehensively evaluates different paddling techniques and their impact on user experience, physical exertion, and motion sickness.
The paper has received mixed feedback from the reviewers and is at a borderline decision as they identified various strengths and weaknesses. The reviewers provided balanced feedback, with strengths in the innovative approach and detailed evaluation but weaknesses in cultural inclusiveness, research novelty, and participant diversity. However, improvements should be made to address the limitations highlighted by the reviewers. I don't think the authors have enough time to address these concerns and produce a paper ready for publication by the conference deadline.

Several common limitations have been identified:
1. Cultural Accuracy: The digital twin uses two oars, which deviates from the traditional single-side paddling technique of dragon boat racing. This discrepancy affects the authenticity of the cultural representation. (Reviewer 3QWC, Reviewer ZBj7)
2. Research Novelty: Reviewers have noted a lack of a novel research question and pointed out existing literature on VR rowing simulations that the authors seemed unaware of. Including references to relevant prior work and highlighting specific innovations in MetaDragonBoat would strengthen the paper. (Reviewer yV5P, Reviewer eeU5)
3. Participant Diversity: The study's findings are limited by a small and homogeneous participant pool, primarily young adults. Including a more diverse demographic would improve the generalizability of the results. (Reviewer ZBj7)
4. Motion Sickness and Fatigue: The evaluation period was short, and longer-term studies are needed to better understand the impact of physical exertion and motion sickness on user experience. (Reviewer ZBj7)
5. Terminology Clarification: The paper uses "VR" and "Metaverse" interchangeably, which can be confusing. Clarifying these terms would enhance the clarity of the discussion. (Reviewer yV5P)

In addition, although the authors have argued their work is motivated by cultural preservation (paper as well as in the rebuttal), neither their study nor the literature review discussed the connection or the contribution towards preserving culture. This is also in accordance with Reviewer eeU5's comment "….would have suggested perhaps a different focus for the paper".

Based on the reviews, the paper is recommended for weak rejection. While the paper presents an interesting and innovative use of VR for cultural heritage preservation, several critical shortcomings must be addressed. The paper requires substantial revisions and a more rigorous approach to be suitable for acceptance. I would strongly encourage authors to present this research as a poster.